# Effect of Ascites Syndrome on Diversity of Cecal Microbiota of Broiler Chickens

**DOI:** 10.3390/vetsci12020126

**Published:** 2025-02-05

**Authors:** Aikebaier Reheman, Zhichao Wang, Ruihuan Gao, Jiang He, Juncheng Huang, Changqing Shi, Meng Qi, Xinwei Feng

**Affiliations:** 1College of Animal Science and Technology, Tarim University, Alar 843300, China; akpar0902@163.com (A.R.); qimengdz@163.com (M.Q.); 2Engineering Laboratory of Tarim Animal Diseases Diagnosis and Control, Xinjiang Production & Construction Corps, Alar 843300, China; 13177890078@163.com (Z.W.); gaoruihuan0@163.com (R.G.); 17591556804@163.com (J.H.); 3Key Laboratory of Livestock and Forage Resources Utilization around Tarim, Ministry of Agriculture and Rural Affairs, Tarim University, Alar 843300, China; hjcll20001020@163.com (J.H.); scq926@163.com (C.S.)

**Keywords:** ascites syndrome, cecal microorganisms, 16S rDNA, broilers

## Abstract

Ascites syndrome (AS) is a metabolic disease that impacts the growth and development of broiler chickens. The intestinal microbiota is closely associated with the growth and development of broiler chickens. However, the effects of AS on the intestinal microbiota of broiler chickens have not yet been reported. The purpose of this study was to demonstrate whether AS affects the intestinal microbiota of broiler chickens. This study showed that AS significantly increased the abundance of *Bacteroidetes* while reducing the abundance of *Actinobacteria*. This study provides data for the in-depth study of the relationship between AS and intestinal microbiota.

## 1. Introduction

Broiler ascites syndrome (AS), also known as broiler pulmonary hypertension syndrome (PHS), is a common nutritional and metabolic disease in rapidly growing commercial broilers [1]. Many studies have shown that diet, management, environment, genetics, and other factors are related to the occurrence of AS; among them, low-temperature, high-energy, high-protein, and high-salt diets are the main feeding and management factors leading to AS [2]. The above factors lead to a series of pathophysiological changes in chickens due to hypoxia, such as increased pulmonary artery blood pressure, pulmonary artery remodeling, right ventricular hypertrophy and failure, free radical production, and increased lipid peroxidation [3]. The above shows that hypoxia is the main cause of AS in broilers.

A hypoxic environment can lead to a significant reduction in the intestinal microbial diversity of broiler chickens. Research indicates that the proportion of certain beneficial bacteria, such as lactic acid bacteria, increases under anoxic conditions, while the abundance of other bacteria, including specific anaerobic species, decreases [4,5]. This shift in microbial composition may adversely affect the health and growth of broilers, impairing their immune function and digestive capacity. Additionally, studies have demonstrated that hypoxia diminishes the growth and feed conversion rates of broiler chickens by impairing the utilization efficiency of specific nutrients by microorganisms [6]. This, in turn, adversely impacts the lipid metabolism, growth performance, and overall health status of broiler chickens [7]. Research shows a potential relationship between the occurrence of PHS and eight specific gut microbiota [8]. There is evidence that PHS is associated with increased gut–hypothalamic paraventricular nucleus transport, pathological changes in the gut wall, and gut microbiota imbalance [9]. Based on the above, we speculate that AS may affect the composition of intestinal microbiota.

Gut microbiota refers to the diverse community of microorganisms, including bacteria, fungi, viruses, and protozoa, that inhabit the intestines of animals. These microorganisms play a crucial role in digestion, immune function, nutrient absorption, and overall health. Many studies have shown that the composition and diversity of microorganisms in the intestines of broiler chickens directly affect their growth rate, feed conversion rate, and disease resistance [10,11,12]. Numerous studies have demonstrated that intestinal microorganisms generate a diverse array of metabolites that are advantageous to the host during metabolic processes, including short-chain fatty acids (SCFAs), vitamins, and other bioactive compounds [13,14]. However, metabolic diseases may alter the gut microbiome and thus negatively impact broiler chickens and disrupt the production of these beneficial metabolites [15,16].

In summary, AS, as a common metabolic disease in broiler chickens, may also affect the composition of the intestinal microbiota of broiler chickens. Therefore, to study the impact of AS on the intestinal flora of broiler chickens, this article analyzed the intestinal microbial diversity of AS broiler chickens by constructing a broiler ascites syndrome model and analyzing intestinal microbial flora.

## 2. Materials and Methods

### 2.1. Ethical Statement

The laboratory animals were housed and treated following the Regulations for the Administration of Affairs Concerning Experimental Animals of the People’s Republic of China and following the Research Ethics Committee of Tarim University (2023025).

### 2.2. Study Design and Treatments

A group of 260 healthy broiler chickens were selected at 0 days old, male in gender and Arbor Acres in breed, and raised intensively in single-layer cages in Alar, China, at an altitude of about 1011 m. Daily management adhered to the routine feeding and management technical specifications for broiler chicks, providing free access to feed and water, artificial lighting throughout the day, and daily cleaning of feces trays to maintain a hygienic environment within the chicken house, and weighing occurred on days 0, 18, and 28. The diet ingredients and nutrients provided during the experiment are shown in Table 1. After 7 days of normal feeding, the feed was withheld for 12 h, and the chicks were subsequently weighed. A total of 240 chicks, all of equal weight and in good mental condition, were selected and randomly divided into four groups, each containing three replicates with 20 chicks per replicate. The mental condition of broiler chickens is primarily assessed by evaluating their levels of exercise, food intake, and water consumption. Signs of a good mental state in broiler chickens include quick movement, acute hearing, bright and lively eyes, long and loud vocalization, a strong appetite, and responsiveness to external stimuli. The experimental group received drinking water with sodium chloride concentrations of 0.1%, 0.2%, and 0.3%, respectively. In contrast, the control group was provided with drinking water that contained no sodium chloride, while both groups were maintained under identical feeding conditions. The broiler chickens were raised until 28 days of age, and the ascites rate, body weight, and ascites heart index (AHI) were calculated. The formula for calculating the AHI is as follows: AHI = right ventricular mass/total ventricular mass. If the peritoneal effusion exceeds 20 mL and the ascites heart index is greater than 0.25, the presence of either of the aforementioned conditions can be classified as broiler ascites syndrome.

### 2.3. Sample Collection

The broiler chickens were euthanized under anesthesia after being raised for 28 days to assess the rate of ascites. Three chickens were randomly selected from the control group and 0.2% sodium chloride group, and the cecal contents were cryopreserved for the extraction of intestinal microbial genomes.

### 2.4. Microbial Total DNA Extraction

Total microbial DNA was extracted from broiler chicken cecal contents according to the protocol described previously [17]. Microbial genomic DNA was isolated with a commercial kit (catalog no. 51504; Qiagen, Shanghai, China; Shanghai Limin Industrial Co., Ltd., Shanghai, China) according to the manufacturer’s description. Prior to high-throughput sequencing, all DNA samples underwent testing via 1.5% agarose gel electrophoresis to verify their integrity and quality. Additionally, the purity of the DNA was assessed using a 2000 NanoDrop (Thermo Fisher Scientific, Waltham, CA, USA) to ensure the reliability of subsequent experiments.

### 2.5. 16S rRNA Sequencing

The amplification of microbial DNA was carried out using specific barcode primers targeting the v3-v4 region of the 16S rRNA gene; the PCR primer sequence was F-5′-ACTCCTACGGGAGGCAGCA-3′), (R-5′-GGACTACHVGGGTWTCTAAT-3′). A commercial PCR kit (New England Biolabs, Ipswich, MA, USA) was used for PCR amplification. Amplifiers were quantified on a 1.5% agarose gel, and light bars between 400 and 450 bp were chosen for sequencing. The libraries were constructed using the NEB Next Ultra DNA Library Pre^®^Kit for Illumina, index codes were added, and the library was quantified using Qubit and Q-PCR. After library identification, computational sequencing was performed using NovaSeq 6000.

### 2.6. Bioinformatic Analysis

As per the quality control procedures outlined in QIIME (Version 1.7.0) (http://qiime.org/scripts/split_libraries_fastq.html) accessed on 21 July 2024 [18], the original tags were quality-filtered under certain filtering conditions to obtain high-quality clean tags [19]. OTU clustering and taxonomic annotation sequence analysis were performed on all valid tags using UPARSE software (Version 20.0.0.36) accessed on 24 July 2024 [20], and sequences with a similarity of ≥97% were assigned to the same OUT. For every representative sequence, the SSU rRNA database from the SILVA138 repository (Version SILVA 138.1) (http://www.arb-silva.de/, accessed on 1 February 2025) was utilized accessed on 10 August 2024 [21] and annotated using the parental method, and species annotation was performed at each taxonomic level (threshold: 0.8–1) [22]. To efficiently analyze various sequences, we conducted phylogenetic diversity and linkage assessments on all representative OTU sequences utilizing MUSCLE software (Version 3.8.31) [23]. The abundance information of OTUs was standardized by utilizing the default sequence count related to the sample with the fewest sequences. Subsequent alpha and beta diversity assessments were conducted on the standardized data. All computations were executed in QIIME (version 1.7.0) and represented visually using R software (version 2.15.3).

### 2.7. Statistical Analysis

The experimental data were analyzed using GraphPad Prism 8.4.3. Student’s *t*-test was used to compare the mean values between the control group and the experimental group. If *p* < 0.05, the difference was considered significant. Experimental results are expressed as mean ± SEM.

## 3. Results

### 3.1. Construction of AS Model Induced by High-Salt Diet

To induce AS, sodium chloride concentrations of 0.1%, 0.2%, and 0.3% were added to the drinking water of broiler chickens beginning at 8 days of age. At 28 days of age, the broiler chickens were euthanized under anesthesia, and their growth performance and cecal microbial diversity were assessed (Figure 1A). There was no significant difference in the body weight of the 28-day-old chickens between the 0.1% sodium chloride group and the control group. However, the addition of 0.2% and 0.3% sodium chloride to the broilers’ drinking water significantly impacted the weight gain of broiler chickens. The results indicated that the 0.2% sodium chloride group experienced a weight reduction of 22.57% compared to the control group. Furthermore, the 0.3% sodium chloride group demonstrated a more pronounced weight reduction of 37.10% (Figure 1B). This study further analyzed the incidence of ascites, revealing that the addition of 0.2% and 0.3% sodium chloride to drinking water resulted in ascites rates of 60% and 70%, respectively. In contrast, the inclusion of 0.1% sodium chloride did not induce ascites in broiler chickens (Figure 1C). Finally, the ascites heart index (AHI) was evaluated, revealing that 0.2% and 0.3% sodium chloride significantly increased the AHI compared to the control group. In contrast, 0.1% sodium chloride did not induce any changes in the broilers (Figure 1D). Therefore, when we subsequently analyzed the impact of AS on intestinal microorganisms, we used the 0.2% sodium chloride group as the experimental group.

### 3.2. AS Reduces Cecal Microbial Diversity in Broiler Chickens

The rarefaction curves demonstrated a trend toward saturation, suggesting that at the 97% similarity threshold, the coverage of microbiome sequences in the two sample groups was adequate to assess the richness and diversity of the microbial community. The Chao index is a diversity estimation metric that relies on the abundance distribution of individual species within a sample. The Shannon index, grounded in information theory, serves as a measurement of species richness and evenness in an ecosystem. Faith’s phylogenetic diversity (Faith-pd) is a diversity index derived from phylogenetic trees. Consequently, this study employed these three indices to analyze the rarefaction curve. The findings indicate that the rarefaction curve levels off and approaches stability, suggesting that the quantity of OTUs acquired for every sample is adequate (Figure 2A). To analyze the effects of AS on the microbial community, principal component analysis (PCA) and principal coordinate analysis (PCoA) were employed to assess the distribution of microorganisms between the two groups. The results showed that the composition of cecal microorganisms was significantly different between the AS group and the control group (Figure 2B). The differences in cecal microbiota abundance between the two groups were further analyzed, and the Venn results indicated that the AS group contained 2198 OTUs, while the control group had 3110 OTUs. Notably, there was a total of 542 OTUs shared between the two groups (Figure 2C). The above results indicate that AS significantly reduces the diversity of broiler cecal microbiota.

### 3.3. AS Reduces Alpha Diversity of Broiler Cecal Microbiota

Alpha diversity represents species richness, evenness, and diversity. This study selected the Shannon, Simpson, and Pielou-e indices for diversity analysis. The results showed that the alpha diversity of the cecal microbiota of the broiler chickens in the AS group was significantly reduced compared with that of the control group (Figure 3A–C).

### 3.4. AS Affects the Composition of Cecal Microorganisms in Broiler Chickens

Changes in the composition of intestinal microbiota are among the primary causes of intestinal microbiota imbalance. Consequently, this study further investigated the effect of AS on the composition of cecal microorganisms in broiler chickens. Histogram analysis showed significant differences in the taxonomic composition of the cecal microorganisms when comparing the AS group to the control group at both the phylum (Figure 4A) and genus (Figure 4B) levels. The results indicated that Bacteroidetes and Firmicutes were the predominant bacterial phyla in the microbial communities of both the AS and control groups, with *Barnesiella* and *Lactobacillus* identified as the dominant genera. Further, LEfSe analysis was performed, and the results showed that Bacteroidetes was significantly increased in the AS group, while Actinobacteria was significantly decreased (Figure 4C). In addition, the abundance of *Dysgonomonas* and *Burkholderia* in the cecum of AS broiler chickens at the genus level was significantly increased, while the abundance of *Melissococcus*, *Oscillospira*, *Ruminococcus*, *Lysobacter*, *Christensenella*, *Anaerofustis*, *Slackia*, and *Olsenella* was significantly decreased (Figure 4D). This shows that AS significantly affects the composition of cecal microorganisms in broiler chickens.

### 3.5. Differential Metabolic Pathway Analysis

Based on the PICRUSt2 (Phylogenetic Investigation of Communities by Reconstruction of Unobserved States) differential microbiota function prediction method, the differential metabolic pathways of the two groups of cecal flora were analyzed. The results showed that the metabolites of AS broiler cecal microorganisms in novobiocin biosynthesis, polycyclic aromatic hydrocarbon degradation, the phosphotransferase system (PTS), bisphenol degradation, and linoleic acid metabolism pathways were significantly reduced, while the metabolites in beta-alanine metabolism, ubiquinone and another terpenoid–quinone biosynthesis, polyketide sugar unit biosynthesis, sphingolipid metabolism, and non-homologous end-joining pathways were significantly increased, especially the metabolites in novobiocin biosynthesis and non-homologous end-joining pathways (Figure 5). The above results indicate that AS affects intestinal health by changing the composition of the broiler cecal microbiota and affecting its metabolites.

## 4. Discussion

AS is a metabolic disease of broiler chickens caused by many factors, which seriously affect the growth and development of broiler chickens and cause significant economic losses in broiler chicken production [24]. Studies have shown that AS changes the normal histological structure of the heart, lungs, and liver of broiler chickens, thereby affecting their growth and development [25,26]. The intestinal microbiota, as the main component of the intestine, regulates normal digestion and absorption function [27]. However, the impact of AS on the intestinal health of broiler chickens, especially on the intestinal microbiota, requires further study.

To investigate the impact of AS on the composition of intestinal microbiota in broiler chickens, this study initially introduced varying concentrations of sodium chloride into the drinking water of the chickens to induce AS. Following this, 16S rRNA sequencing technology was employed to analyze the diversity of the intestinal microbiota. The results showed that sodium chloride concentrations of 0.2% and 0.3% significantly induced AS in broiler chickens. Guo et al. added 0.12% sodium chloride to drinking water to induce ascites syndrome in broiler chickens [28]. Consequently, this study further investigated the impact of AS induced by 0.2% sodium chloride on the cecal microbial diversity of broiler chickens.

The intestinal microbiota helps maintain the physiological structure and function of the intestine and is closely related to intestinal inflammation, barrier function, and host growth performance [29]. The composition of the intestinal microbiota is affected by different factors such as pathological conditions, genetics, environment, age, and diet, which are expected to play a crucial role in growth, immunity, and nutrient absorption [30]. The host diet significantly affects the intestinal microbiota, which in turn affects the host metabolism [31]. Therefore, this study further analyzed the effect of high-salt-diet-induced AS on the cecal intestinal microbiota of broiler chickens.

Numerous studies have shown that abnormal changes in the intestinal microbial community exist in AS patients and various animal models [32]. This study shows that AS significantly reduces the cecal microbial diversity of broiler chickens. Research shows that higher intestinal microbial diversity is beneficial to intestinal health [33]. This shows that AS affects intestinal health by reducing the microbial diversity of the cecum of broiler chickens. Alpha diversity is indicative of the distribution and abundance of microbial populations. The diversity, uniformity, and richness of microorganisms serve as important indicators influencing intestinal health [34]. Recent studies have demonstrated that AS increases the abundance of *Firmicutes*, *Proteobacteria*, *Actinobacteria*, *Firmicutes-Clostridium*, and *Gammaproteo* bacteria in the intestinal tract of rats while concurrently reducing the abundance of *Bacteroidetes*, *Spirochaetes*, *Bacteroidia*, *Spirochaetia*, and *Bacilli* [35]. This study demonstrates that AS reduces the alpha diversity of broiler cecal microorganisms, with the Shannon, Simpson, and Pielou-e indices showing marked reductions compared to the control group. The analysis of the cecal microbial composition found that AS changed the cecal microbial composition of broiler chickens at the phylum and genus levels. The LEfSe analysis results showed that AS reduced the abundance of *Actinobacteria* and increased the abundance of *Bacteroidetes* at the phylum level. There exists a gap between this result and the aforementioned findings, which may be attributed to the varying effects of AS on the intestinal flora across different species. *Firmicutes* and *Bacteroidetes* are the predominant bacterial phyla in the intestines of animals. *Bacteroidetes’* primarily function is to degrade carbohydrates, and their increased abundance is negatively correlated with the daily weight gain of animals [36]. AS may increase the abundance of cecal *Bacteroidetes*, thereby affecting the weight gain of broiler chickens. The intestinal microbiota regulates various physiological functions of the host by synthesizing and secreting specific metabolites [37]. Therefore, this study used bioinformatics prediction to further analyze differential metabolic pathways, and the results showed that AS significantly reduced metabolites related to polycyclic aromatic hydrocarbon degradation pathways. Polycyclic aromatic hydrocarbons are toxic organic pollutants that seriously affect the health of humans and animals, and bacteria have a strong ability to degrade polycyclic aromatic hydrocarbons [38]. It is suggested that AS may reduce bacterial abundance related to the degradation of PAH compounds in the cecum, thereby further affecting the intestinal health of broiler chickens.

This study shows that AS affects the diversity and abundance of broiler cecal microbiota. Thus, the gut microbiome can be used as a target to predict and improve AS in broiler chickens. These findings may provide valuable information for future studies on the impact of AS on intestinal microbial imbalance in broiler chickens. However, further research is needed on the interaction between cecal microorganisms and ascites syndrome.

## 5. Conclusions

In conclusion, this study demonstrated that AS significantly influences the diversity and composition of cecal microorganisms in broilers. This study provides foundational data for subsequent research on the interaction between intestinal microbiota and AS.

## Figures and Tables

**Figure 1 vetsci-12-00126-f001:**
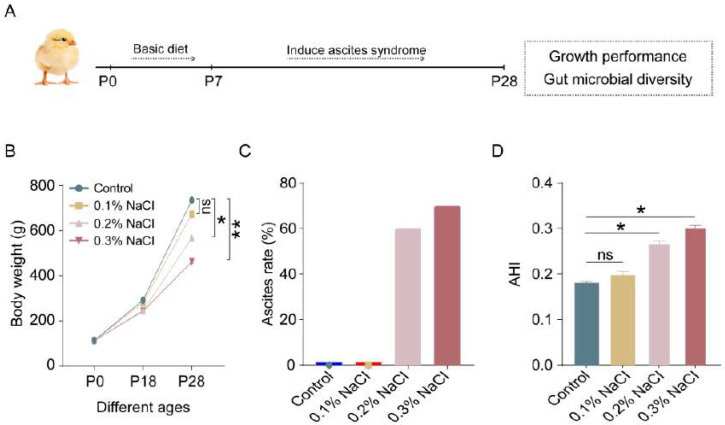
Construction of AS model induced by a high-salt diet. (**A**) Schematic of experimental design pattern. (**B**) Effects of different concentrations of sodium chloride on changes in body weight of broiler chickens. (**C**) Effects of different concentrations of sodium chloride on ascites rate in broiler chickens. (**D**) Effects of different concentrations of sodium chloride on AHI in broiler chickens. The experiment was performed in three biological replicates. All data are presented as mean ± SEM. The probability of significant differences was analyzed using Student’s *t*-test. * *p* ≤ 0.05, ** *p* ≤ 0.01 compared with the control, ns means *p* > 0.05.

**Figure 2 vetsci-12-00126-f002:**
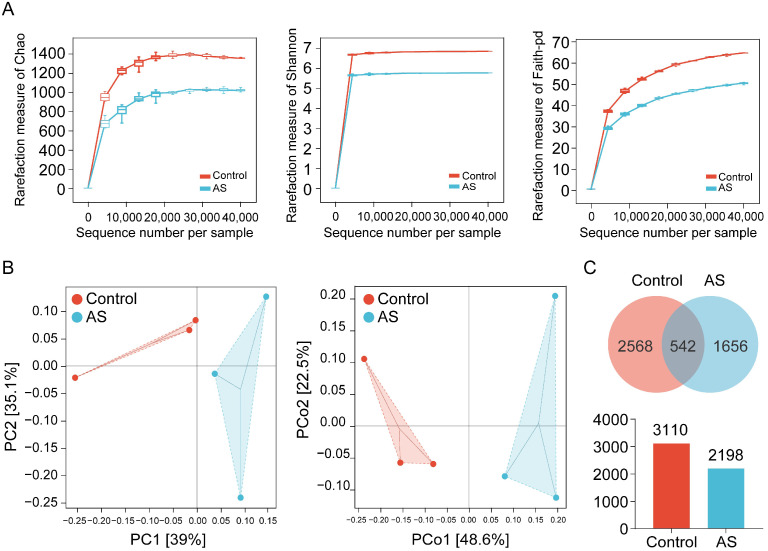
Analysis of the impact of AS on the number of cecal microorganisms. (**A**) Rarefaction curve plot of sequence number per sample group, such as Chao, Shannon, and Faith-pd. (**B**) Genus-level PCA of operational taxonomic units from sample groups and genus-level PCoA plots of microbial community dissimilarity (Bray–Curtis) in each sample-weighted uniFrac. Different colors represent different sample groups. (**C**) Venn diagrams on OTU level.

**Figure 3 vetsci-12-00126-f003:**
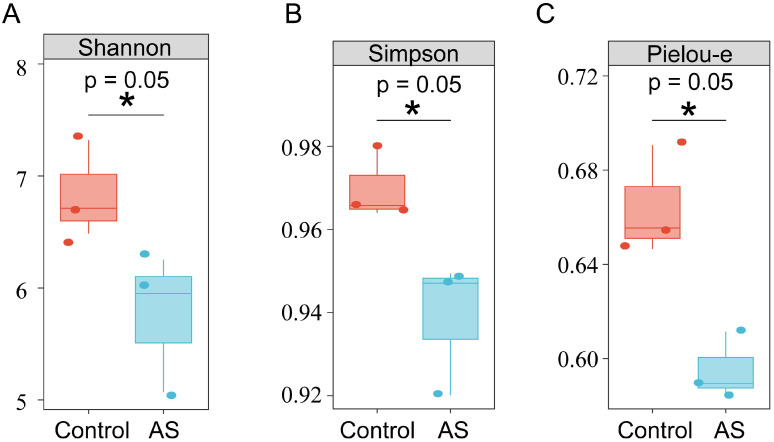
The alpha diversity in cecal microbiota. (**A**) Shannon index on OUT level. (**B**) Simpson index on OUT level. (**C**) Pielou-e index on OUT level. Statistical differences in the average values of each indicator were analyzed through independent *t*-tests. * indicates *p* ≤ 0.05.

**Figure 4 vetsci-12-00126-f004:**
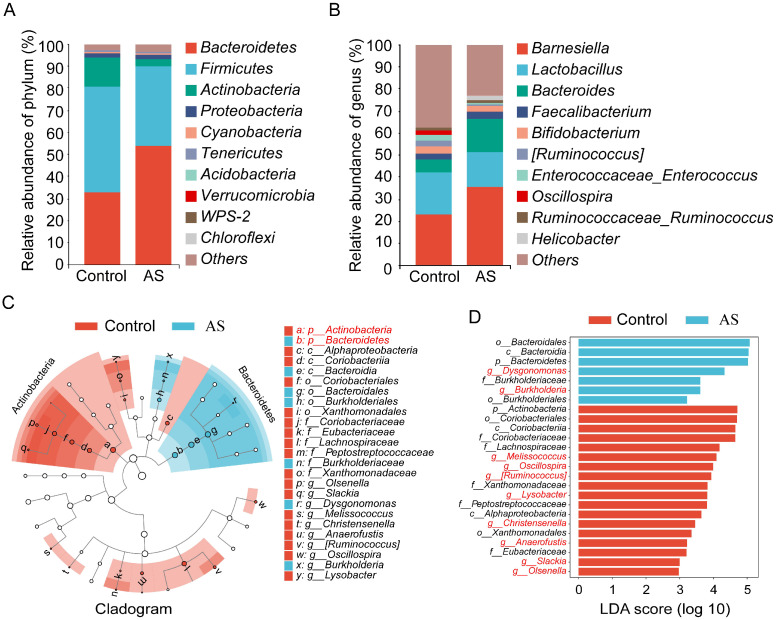
Analysis of the impact of AS on the composition of cecal microbiota. (**A**,**B**) Relative abundance of cecal microbiota at phylum and genus levels. (**C**) Phylogenetic profiles of specific bacterial taxa and dominant bacteria in two distinct groups were determined using LEfSe analysis. Biomarker groups are represented by colored circles and shaded areas. The diameter of each circle depends on the abundance of the taxon in the community. (**D**) Differential microbial enrichment between the control and AS groups is illustrated, with the red and blue horizontal color bars representing the microbial enrichment in the cecum of broiler chickens for the intact and AS groups, respectively.

**Figure 5 vetsci-12-00126-f005:**
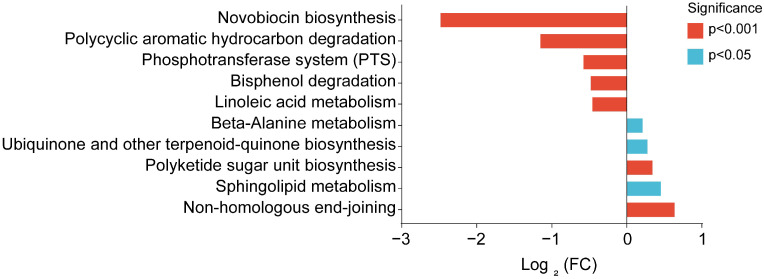
Analysis of differential metabolic pathways in the AS group compared with the control group. FC represents the change in metabolite abundance in the AS group compared with the control group.

**Table 1 vetsci-12-00126-t001:** Basic diet composition and nutrient level.

Ingredients	Content (%)	Nutrient Level	
Maize	59.00	Metabolizable energy/(MJ/kg)	13.56
Soybean meal	27.00	Dry matter (%)	86.00
Wheat bran	5.00	Crude protein (%)	19.58
Rapeseed meal	3.00	Crude fiber (%)	6.00
Calcium hydrogen phosphate	2.50	Coarse ash (%)	7.00
Sunflower meal	1.50	Calcium (%)	0.75
Stone powder	0.35	Total phosphorus (%)	0.45
Premix	1.35	Sulfur-containing amino acids (%)	0.50
Sodium chloride	0.3	Water (%)	0.50
Total	100.00	Total	

1. The premix involves the confidential formula of Xinjiang Tiankang Feed Technology Co., Ltd. (Urumqi, China), which will not be disclosed. 2. The nutritional levels are analytically guaranteed values tested by Xinjiang Tiankang Feed Technology Co., Ltd.

## Data Availability

The raw data supporting the conclusions of this article will be made available by the authors on request.

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
