# Peer review of "Effect of Ascites Syndrome on Diversity of Cecal Microbiota of Broiler Chickens"

_vetsci, 2025, doi:10.3390/vetsci12020126_

Round 1
Reviewer 1 Report
Comments and Suggestions for Authors
Effect of Ascites Syndrome on the Diversity of Cecal Microbiota of Broiler Chickens
Ascites syndrome (AS) in 16 broiler chickens was induced through high salt intake, and the effects of AS on the cecal flora were analyzed using 16S rDNA sequencing technology. The results showed that AS significantly reduced the cecal microbial diversity of broiler chickens and affected the cecal microbial composition at the phylum and genus levels. LEfSe analysis revealed that AS significantly increased the abundance of Bacteroides while reducing the abundance of actinobacteria in the cecum. The authors concluded that AS may further impact the growth rate of broiler chickens by altering the diversity and composition of the intestinal flora in their ceca.
L45: add references
Introduction: you need to add a paragraph about the relationship between AS and microbiota.
L69: d
how did you determine that the birds were healthy?
More information is required about the birds' strain, management, etc.
It's not clear how these treatments induced AS, you need to clarify, why you have selected those levels.
What was the experimental design?
L125: what was the proof? Did the diet have any sodium chloride?
Provide more data about feed intake and FCR.
Revise the discussion and conclusion.
Author Response
Response to Reviewer 1 Comments
|
|||||||||||||||||||||||||||
1. Summary |
|
|
|||||||||||||||||||||||||
Ascites syndrome (AS) in broiler chickens was induced through high salt intake, and the effects of AS on the cecal flora were analyzed using 16S rDNA sequencing technology. The results showed that AS significantly reduced the cecal microbial diversity of broiler chickens and affected the cecal microbial composition at the phylum and genus levels. LEfSe analysis revealed that AS significantly increased the abundance of Bacteroides while reducing the abundance of actinobacteria in the cecum. The authors concluded that AS may further impact the growth rate of broiler chickens by altering the diversity and composition of the intestinal flora in their ceca. Thank you for reviewing our manuscript and providing constructive comments, which greatly helped us improve it. The manuscript has been carefully revised and the point-by-point responses are as follows. We want your comments to be processed accurately. Revised manuscripts are highlighted in yellow and responses are presented in red text. Once again, thank you very much for your critical comments and inspiring suggestions. Best regards, Xin wei Feng |
|||||||||||||||||||||||||||
2. Point-by-point response to Comments and Suggestions for Authors |
|||||||||||||||||||||||||||
Comments 1: Comments 1: L45: add references. |
|||||||||||||||||||||||||||
Response 1: Thank you for pointing this out. I agree with this comment and added references, as shown on line 55 of the manuscript. |
|||||||||||||||||||||||||||
Comments 2: Introduction: you need to add a paragraph about the relationship between AS and microbiota. |
|||||||||||||||||||||||||||
Response 2: Thank you for pointing this out. I agree with this comment and added a paragraph about the relationship between AS and the microbiota, as shown on line 56-60 of the manuscript. |
|||||||||||||||||||||||||||
Comments 3: L69: d. |
|||||||||||||||||||||||||||
Response 3: I would like to express my sincere gratitude to the reviewer for dedicating time from their busy schedule to review my paper. However, I am unclear about the meaning of the comment labeled "d" that you provided. |
|||||||||||||||||||||||||||
Comments 4: how did you determine that the birds were healthy? |
|||||||||||||||||||||||||||
Response 4: Thank you for pointing this out. The chicks utilized in this study consisted of 260 0-day-old Arbor Acres broilers, which were hatched by the our team and the chicks raised in strict adherence to established broiler feeding standards. Throughout this feeding period, we monitored the mental state, as well as the feed and drinking water conditions of the broilers. After 7 days, 240 broiler chickens, each 7 days old and exhibiting similar body weight and optimal body condition, were selected for follow-up research. The origin of these broilers is well documented, and the feeding standards are consistent to ensure their health and full compliance with the requirements of this experiment. Evaluation criteria have been added in line 93-98 of the manuscript. |
|||||||||||||||||||||||||||
Comments 5: More information is required about the birds' strain, management, etc. |
|||||||||||||||||||||||||||
Response 5: Thank you for pointing this out. I agree with this comment and added bird strains and management methods in the manuscript, as shown on line 85-107 of the manuscript. |
|||||||||||||||||||||||||||
Comments 6: It's not clear how these treatments induced AS, you need to clarify, why you have selected those levels. |
|||||||||||||||||||||||||||
Response 6: Thank you for pointing this out. Firstly, reference was made to published literature (Guo et al., 2023), In this study, AS was induced by adding 0.12% sodium chloride to drinking water. In this study, we added three different concentrations of sodium chloride—0.1%, 0.2%, and 0.3%—to the drinking water of broiler chickens. We monitored weight changes, calculated ascites rates, and assessed the ascites heart index (AHI). Our results indicate that the induction effect is most pronounced with the addition of 0.2% and 0.3% sodium chloride to the drinking water. Consequently, we utilized 0.2% sodium chloride to induce AS in broiler chickens in subsequent experiments. |
|||||||||||||||||||||||||||
Comments 7: What was the experimental design? |
|||||||||||||||||||||||||||
Response 7: Thank you for pointing this out. Experimental design is described in the manuscript, as shown on line 85-95 of the manuscript. |
|||||||||||||||||||||||||||
Comments 8: L125: what was the proof? Did the diet have any sodium chloride? |
|||||||||||||||||||||||||||
Response 8: Thanks for pointing this out. Based on your suggestion, I have added basic dietary composition information to the manuscript. The diets used in this study contained 0.3% sodium chloride, as indicated in line 108 of the manuscript. |
|||||||||||||||||||||||||||
Comments 9: Provide more data about feed intake and FCR. |
|||||||||||||||||||||||||||
Response 9: Thank you for pointing this out. I agree with this comment, unfortunately, the data pertaining to feed intake and FCR have been published in other Chinese journals, which prevents us from including this information in the manuscript. We apologize for any inconvenience this may cause. Below, we present our published data on average daily water intake, average daily feed intake, average daily weight gain, and feed-to-weight ratio. Table 1. Data related to feed intake and FCR
|
|||||||||||||||||||||||||||
Comments 10: Revise the discussion and conclusion |
|||||||||||||||||||||||||||
Response 10: Thank you for pointing this out. I agree with this comment and have made detailed revisions to the discussion and conclusion sections of the manuscript as per your suggestions, as shown on line 277-280、291-293、298-310、311-312、314-318、323-324 and 330-332 of the manuscript. |

Reviewer 2 Report
Comments and Suggestions for Authors
The manuscript explores the effect of ascites syndrome on the cecal microbiota of broiler chickens. The study is interesting because ascites syndrome is responsible for significant economic losses in chicken production at altitudes higher than 1,100 meters above sea level. The main weaknesses of the study are observed in the material and methods. The authors do not indicate the genetic line or breed of the broiler chicken used in the experiment; the formulation of the feed that the birds received is not shown, and they do not indicate if the same feed was used throughout the experiment or if there was more than one feeding phase.
The authors mention the selection of birds based on a good condition but do not indicate how that mental condition was estimated. Also, the description of the method for calculating the ascites rate and ascites heart index is missing, as well as specifying the number of birds evaluated for these determinations.
In the results section, Figures 2, 3, and 4 show an "Intact" group, but in the text, it is handled as a control group.
The title of Figure 5 must indicate that it corresponds only to the group of ascites syndrome.
The discussion section needs more contrast between the results of this study and others, as well as the impact or effect that the change in the diversity of the cecal microbiota may have in terms of specific values for performance parameters and not in general. For example, in lines 257-258 it is mentioned that the increase in the abundance of Bacteroidetes is negatively related to daily weight gain, but it does not indicate by how much.
The PDF file contains some notes, corrections, and suggestions for the manuscript.

Author Response
Response to Reviewer 2 Comments
|
||
1. Summary |
|
|
Thank you for reviewing our manuscript and providing constructive comments, which greatly helped us improve it. The manuscript has been carefully revised and the point-by-point responses are as follows. We want your comments to be processed accurately. Revised manuscripts are highlighted in yellow and responses are presented in red text. Once again, thank you very much for your critical comments and inspiring suggestions. Best regards, Xin wei Feng |
||
2. Point-by-point response to Comments and Suggestions for Authors |
||
Comments 1: The manuscript explores the effect of ascites syndrome on the cecal microbiota of broiler chickens. The study is interesting because ascites syndrome is responsible for significant economic losses in chicken production at altitudes higher than 1,100 meters above sea level. The main weaknesses of the study are observed in the material and methods. The authors do not indicate the genetic line or breed of the broiler chicken used in the experiment; the formulation of the feed that the birds received is not shown, and they do not indicate if the same feed was used throughout the experiment or if there was more than one feeding phase. |
||
Response 1: Thanks for pointing this out. I acknowledge this oversight and apologize for not specifying the broiler breed and diet composition in the original manuscript. These details have now been incorporated into the revised manuscript, as presented in table 1, and are referenced on line 85、108. |
||
Comments 2: The authors mention the selection of birds based on a good condition but do not indicate how that mental condition was estimated. Also, the description of the method for calculating the ascites rate and ascites heart index is missing, as well as specifying the number of birds evaluated for these determinations. |
||
Response 2: Thank you for pointing this out. We may have neglected to provide a detailed description of the mental status of broiler chickens, which has been addressed in the revised manuscript, as shown on line 93-98 of the manuscript. Additionally, Since some broiler chickens died during feeding, the last 200 chickens were used for the determination of ascites rate and AHI. The ascites rate and AHI calculation methods have been described in the revised manuscript, as shown in line104-105 of the manuscript. |
||
Comments 3: In the results section, Figures 2, 3, and 4 show an "Intact" group, but in the text, it is handled as a control group. |
||
Response 3:Thank you for pointing this out. I agree with this comment and based on your suggestion, we have changed Intact in all figures and text to the control group. |
||
Comments 4: The title of Figure 5 must indicate that it corresponds only to the group of ascites syndrome. |
||
Response 4: Thank you for pointing this out. I agree with this comment and have been revised in the manuscript, as shown in line 261 of the manuscript. |
||
Comments 5: The discussion section needs more contrast between the results of this study and others, as well as the impact or effect that the change in the diversity of the cecal microbiota may have in terms of specific values for performance parameters and not in general. For example, in lines 257-258 it is mentioned that the increase in the abundance of Bacteroidetes is negatively related to daily weight gain, but it does not indicate by how much. |
||
Response 5: Thank you for pointing this out. I agree with this comment and the discussion section has been revised in detail as per your suggestions, as shown in line 277-280、291-293、298-310、311-312、314-318 and 323-324 of the manuscript. I am very sorry that there is no specific data in this article regarding the relationship between Bacteroidetes and daily weight gain. |
||
Comments 6: The PDF file contains some notes, corrections, and suggestions for the manuscript. |
||
Response 6: thank you for pointing this out. I agree with this comment, according to your suggestions, I have made detailed modifications to the manuscript. |

Reviewer 3 Report
Comments and Suggestions for Authors
The authors should demonstrate with a literature review in the DISCUSSION section that there is evidence that chickens naturally affected with ascites also have modified intestinal microbiota. Otherwise, the work has a confounding effect, since it is not known whether sodium chloride or induced ascites modified the intestinal microbiota.
It is recommended to update the cited literature using citations no older than five years

Author Response
Response to Reviewer 3 Comments
|
||
1. Summary |
|
|
Thank you for reviewing our manuscript and providing constructive comments, which greatly helped us improve it. The manuscript has been carefully revised and the point-by-point responses are as follows. We want your comments to be processed accurately. Revised manuscripts are highlighted in yellow and responses are presented in red text. Once again, thank you very much for your critical comments and inspiring suggestions. Best regards, Xin wei Feng |
||
2. Point-by-point response to Comments and Suggestions for Authors |
||
Comments 1: The authors should demonstrate with a literature review in the DISCUSSION section that there is evidence that chickens naturally affected with ascites also have modified intestinal microbiota. Otherwise, the work has a confounding effect, since it is not known whether sodium chloride or induced ascites modified the intestinal microbiota. |
||
Response 1: Thank you for pointing this out. I agree with this comment and unfortunately, we found no articles on the impact of naturally occurring AS on the intestinal microbiota of chickens. However, we found that studies have reported abnormal changes in the intestinal microbiota in human patients with ascites and other animal models, as shown on line 291-292 of the manuscript. Study shows high-salt diet affects gut microbiota in mice, especially lactobacilli (Chen et al., 2017). However, in this study, all animals were fed the same diet. Both the control group and the experimental group contained 0.3% sodium chloride in their diets. The experimental results showed that the sodium chloride in the diet was not enough to cause AS. However, after different concentrations of sodium chloride were added to the drinking water of the experimental group, AS was induced by 0.2% and 0.3% sodium chloride. Finally, we compared the changes in intestinal microbiota between the experimental group (0.2% sodium chloride) and the control group. Therefore, the difference in intestinal microbiota is caused by AS, not the sodium chloride in the feed. |
||
Comments 2: It is recommended to update the cited literature using citations no older than five years. |
||
Response 2: Thanks for pointing this out. I agree with this comment and some documents have been updated according to your suggestions, but some documents have not been updated because they are very classic. |
||
|
||

Reviewer 4 Report
Comments and Suggestions for Authors
Intro: Recommendation to include more specific details about the factors that can induce AS should be included, and justification on why the 'high-salt' diet was selected as the best trigger for this investigation. The intro currently states 'The above shows that hypoxia is the main cause of AS in broilers'. Does high-salt induce hypoxia? What effect does high-salt have on intestinal microbiota? How will other sources of salt (basal diet + drinking water) be considered or assessed in terms of the total salt intake? What are the side-effects of high-salt diets and how will these be monitored to ensure they don't bias the study (or are considered with analysis of the results)?
Methods: recommendation to include more information about the study site altitude (high altitude has been shown to have an effect on AS incidence); breed (this is mentioned as a risk factor in the intro, so the breed should be specified in the methods); health status, particularly vaccination for infectious bronchitis (did this occur and if so, with what strain); groupings (was each group of 20 broilers housed in a single cage?); diet (were they phase feed with Starter, Grower, Finisher, and what was the salt content of each diet?). Also recommend including more information on how AS was assessed at 28d during necropsy? Was this a published method and if so, please include the reference. Details of growth rate should be included as this is also a correlating factor for AS. Were all birds in each treatment group combined based upon treatment, or how were birds with/ without AS separated within a treatment group for analysis? Was the sex confirmed at necropsy, and if there were any mis-sex chicks, how were these analysed? How was the AHI calculated, and if published, please include the reference. What was the water source, and was this tested for basal salt levels? Were any anticoccidials used in the feed (particularly ionophores which have an effect on the intestinal microbiome)? Does 'equal weight' mean they fell within say 10% of the mean? Does 'good mental condition' mean physical condition? Does 'three chickens from each group were randomly selected' mean only 3 per treatment, or 3 per replicate (i.e. a total of 12 or 36 samples were collected for cecal microbiome analysis)?
Author Response
Response to Reviewer 4 Comments
|
|||||||||||||||||||||||||||
1. Summary |
|
|
|||||||||||||||||||||||||
Thank you for reviewing our manuscript and providing constructive comments, which greatly helped us improve it. The manuscript has been carefully revised and the point-by-point responses are as follows. We want your comments to be processed accurately. Revised manuscripts are highlighted in yellow and responses are presented in red text. Once again, thank you very much for your critical comments and inspiring suggestions. Best regards, Xin wei Feng |
|||||||||||||||||||||||||||
2. Point-by-point response to Comments and Suggestions for Authors |
|||||||||||||||||||||||||||
Comments 1: Intro: Recommendation to include more specific details about the factors that can induce AS should be included, and justification on why the 'high-salt' diet was selected as the best trigger for this investigation. The intro currently states 'The above shows that hypoxia is the main cause of AS in broilers'. Does high-salt induce hypoxia? What effect does high-salt have on intestinal microbiota? How will other sources of salt (basal diet + drinking water) be considered or assessed in terms of the total salt intake? What are the side-effects of high-salt diets and how will these be monitored to ensure they don't bias the study (or are considered with analysis of the results)? |
|||||||||||||||||||||||||||
Response 1: Thank you for pointing this out. I agree with this comment and we have made detailed revisions to the manuscript according to your suggestions, which are shown in yellow. Firstly, feeding and management are the main human factors causing AS, such as low temperature, high energy, high protein, and high salt diet (Yu et al., 2023). Again, referring to the published literature (Guo et al., 2023), this study induced AS by adding 0.12% sodium chloride to drinking water. Therefore, in this study, we used high salt to induce the production of AS to study the impact of AS on intestinal microorganisms. Studies have demonstrated that low-temperature, high-energy, high-protein, and high-salt diets elevate the metabolic rate and tissue oxygen consumption in broilers, resulting in relative hypoxia (Julian, 1993; Wideman et al., 2013). This hypoxia can lead to tissue oxygen free radical damage, increased cardiac output, insufficient pulmonary blood volume, right heart hypertrophy, pulmonary arterial hypertension (PAH), and other related symptoms. PAH represents a central event in broiler AS (Yu et al., 2023). Therefore, although high-salt diet cannot directly cause AS, it indirectly causes AS in broiler chickens by producing tissue hypoxia. Study shows high-salt diet affects gut microbiota in mice, especially lactobacilli (Chen et al., 2017). However, in this study, all animals were fed the same diet. Both the control group and the experimental group contained 0.3% sodium chloride in their diets. The experimental results showed that the sodium chloride in the diet was not enough to cause AS. However, after different concentrations of sodium chloride were added to the drinking water of the experimental group, AS was induced by 0.2% and 0.3% sodium chloride. Finally, we compared the changes in intestinal microbiota between the experimental group (0.2% sodium chloride) and the control group. Therefore, the difference in intestinal microbiota is caused by AS, not the sodium chloride in the feed. Feeding high-salt diet disrupts the electrolyte balance of chickens and affects their physiological functions. The diet utilized in this study is sourced from the Xinjiang Tiankang Company in China. The sodium chloride content of the diet is 0.3%, which falls within the normal range and is not considered high, as shown on line 108 of the manuscript. |
|||||||||||||||||||||||||||
Comments 2: Methods: recommendation to include more information about the study site altitude (high altitude has been shown to have an effect on AS incidence); breed (this is mentioned as a risk factor in the intro, so the breed should be specified in the methods); health status, particularly vaccination for infectious bronchitis (did this occur and if so, with what strain); groupings (was each group of 20 broilers housed in a single cage?); diet (were they phase feed with Starter, Grower, Finisher, and what was the salt content of each diet?). Also recommend including more information on how AS was assessed at 28d during necropsy? Was this a published method and if so, please include the reference. Details of growth rate should be included as this is also a correlating factor for AS. Were all birds in each treatment group combined based upon treatment, or how were birds with/ without AS separated within a treatment group for analysis? Was the sex confirmed at necropsy, and if there were any mis-sex chicks, how were these analysed? How was the AHI calculated, and if published, please include the reference. What was the water source, and was this tested for basal salt levels? Were any anticoccidials used in the feed (particularly ionophores which have an effect on the intestinal microbiome)? Does 'equal weight' mean they fell within say 10% of the mean? Does 'good mental condition' mean physical condition? Does 'three chickens from each group were randomly selected' mean only 3 per treatment, or 3 per replicate (i.e. a total of 12 or 36 samples were collected for cecal microbiome analysis)?. |
|||||||||||||||||||||||||||
Response 2: Thanks for pointing this out. I agree with this comment and and revised the manuscript. I will give you a point-to-point reply below: (1) Altitude information added on line 87 of the manuscript. (2) Chicken breed information added on line 85 of the manuscript. (3) The chickens utilized in this study were hatched in our laboratory and are in good health. The criteria for evaluating the health index have been included in line 93-99 of the manuscript. No infectious diseases. (4) Yes, each group of 20 chickens is placed in a large cage with enough space for movement. The breeding space complies with relevant animal welfare regulations. (5) The experimental period of this study was 28 days. The same diet was used throughout the experiment, which came from Xinjiang Tiankang Company in China and contained 0.3% sodium chloride, as shown on line 108 of the manuscript. (6) The criteria for assessing AS have been added in line 102-107 of the manuscript, which is a published method, published in a Chinese journal (Gu, 2019, Livestock and fishery). (7) Thank you for pointing this out. I agree with this comment, unfortunately, the data pertaining to feed intake and FCR have been published in other Chinese journals, which prevents us from including this information in the manuscript. We apologize for any inconvenience this may cause. Below, we present our published data on average daily water intake, average daily feed intake, average daily weight gain, and feed-to-weight ratio. Table 1. Data related to feed intake and FCR
(8) On the eighth day, we divided the broilers into four groups: a control group, a 0.1% sodium chloride group, a 0.2% sodium chloride group, and a 0.3% sodium chloride group. The broilers were allocated into 12 cages, with 20 animals in each cage, and were raised until they reached 28 days of age. Following necropsy, only the broilers exhibiting AS were selected for the experimental group, while healthy broilers were excluded from this study. The sex of the chick is determined as male by anal inversion after hatching. (9) The AHI calculation method has been added in line 104 of the manuscript,which is a published method, published in a Chinese journal (Gu, 2019, Livestock and fishery). The water consumed by all animals in this study was distilled water and did not contain salt. (10) No anticoccidials are included in the feed. (11) Yes, broilers having the same equal weight means they are within 10% of the mean. (12) In this study, we mainly judge the health status of broiler chickens according to their mental status, and the evaluation method has been added in line 93-99 of the manuscript. (13) When assessing body weight changes, ascites rates, and AHI, more than three animals were used, with at least 50 animals per group (note that some animals died during the experiment). However, when testing intestinal microorganisms, 3 animals each were tested in the control group and 0.2% chloride group. Because both 0.2% and 0.3% sodium chloride can cause AS, so we chose the 0.2% sodium chloride group for subsequent intestinal microbial analysis. |
|||||||||||||||||||||||||||
|
|||||||||||||||||||||||||||

Round 2
Reviewer 1 Report
Comments and Suggestions for Authors
Effect of Ascites Syndrome on the Diversity of Cecal Microbiota of Broiler Chickens Revised version
L20: start with Ascites syndrome (AS).
Include the experimental design in the abstract and P values for significant terms.
Introduction: Include some studies about the relationship between AS and Microbiota. Your work must be justified and the objectives clarified.
Table 1: the analysis should include Na and chloride.
Other parts look good.
Author Response
Response to Reviewer 1 Comments Round 2 |
||
1. Summary |
|
|
Thank you for reviewing our manuscript and providing constructive comments, which greatly helped us improve it. The manuscript has been carefully revised and the point-by-point responses are as follows. We want your comments to be processed accurately. Revised manuscripts are highlighted in yellow and responses are presented in red text. Once again, thank you very much for your critical comments and inspiring suggestions. Best regards, Xin wei Feng |
||
2. Point-by-point response to Comments and Suggestions for Authors |
||
Comments 1: L20: start with Ascites syndrome (AS). |
||
Response 1: Thank you for pointing this out. We have modified the manuscript, as shown on line 21 of the manuscript. |
||
Comments 2: Include the experimental design in the abstract and P values for significant terms. |
||
Response 2: Thank you for pointing this out. Following the reviewer's suggestion, we have added experimental design and P-value in the abstract section of the manuscript, as shown on line 24-30 of the manuscript. |
||
Comments 3: Introduction: Include some studies about the relationship between AS and Microbiota. Your work must be justified and the objectives clarified. |
||
Response 3: There is currently no research report on whether AS affects gut microbiota, but studies have shown that pulmonary hypertension syndrome (PHS) affects gut microbiota, and PHS and AS are actually very similar symptoms. Therefore, we speculate that AS may also affect cecal microbiota. This section has been mentioned in the introduction, as shown on line 55-60 of the manuscript. |
||
Comments 4: Table 1: the analysis should include Na and chloride. |
||
Response 4: Thank you very much for the reviewer's suggestions. We also hope to separately specify the content of Na and chloride. Unfortunately, the company only provided the content of sodium chloride in the diet and did not separately provide the content of Na and chloride. |

Reviewer 4 Report
Comments and Suggestions for Authors
Well done on the additional information included to address the deficiencies, particularly the methods section!
Typo's noted in line 81 - 'Group of...'; and 166 - ascites
Author Response
Response to Reviewer 4 Comments Round 2 |
||
1. Summary |
|
|
Thank you for reviewing our manuscript and providing constructive comments, which greatly helped us improve it. The manuscript has been carefully revised and the point-by-point responses are as follows. We want your comments to be processed accurately. Revised manuscripts are highlighted in yellow and responses are presented in red text. Once again, thank you very much for your critical comments and inspiring suggestions. Best regards, Xin wei Feng |
||
2. Point-by-point response to Comments and Suggestions for Authors |
||
Comments 1: Typo's noted in line 81 - 'Group of...'; and 166 - ascites |
||
Response 1: Thank you for pointing this out. We have corrected, as shown on line 84-169 and of the manuscript. |
||
